# Interoperable Open Specifications Framework for the Implementation of Standardized Urban Platforms [note 1]

**DOI:** 10.3390/s20082402

**Published:** 2020-04-23

**Authors:** José L. Hernández, Rubén García, Joachim Schonowski, Daniel Atlan, Guillaume Chanson, Timo Ruohomäki

**Affiliations:** 1CARTIF Foundation—Energy Division, Parque Tecnológico Boecillo, 205, 47151 Boecillo, Spain; 2Freelancer (Avoid Reduce Reuse)—Digital Expert on Smart Cities (T-Labs during mySMARTLife), Rietze Strasse, 10409 Berlin, Germany; joachimschonowski@gmx.de; 3ENGIE INEO Solutions Digitales, Rue Henri Farman, 86, 92130 Issy-les-Moulineaux, France; daniel.atlan@engie.com; 4Nantes Métropole, Cours du Champ de Mars, 2, 44923 Nantes, France; Guillaume.CHANSON@nantesmetropole.fr; 5Forum Virium Helsinki, Unioninkatu, 24, 00130 Helsinki, Finland; timo.ruohomaki@forumvirium.fi

**Keywords:** smart city, urban data platform, interoperability, open data, open specifications framework, open standards, smart services

## Abstract

The current cities’ urban challenges go through digitalization and integration of new technologies under the perspective of actual and future ecological, as well as socio-economic commitments. This process is translated into the Open Standardized Urban Data Platform, which plays a pivotal role. Within its main functionalities, data ingestion, analytics and services as vertical domains become necessary to create more environmentally friendly cities. However, there still exist some deficits. Among them, openness and interoperability are outlined. On the one hand, there is a lack of open data initiatives for increasing the smart services stock. On the other hand, interoperability depends upon vendors and integrators, reducing the possibilities of Smart City growth. In this context, under the mySMARTLife project (GA #731297) umbrella, an Open Specifications Framework has been developed in order to address four main issues: (1) data interoperability; (2) services or verticals interoperability; (3) openness; and (4) replicability. It enlightens the implementation and integration of multiple city domains (like infrastructures, mobility, energy, buildings) for smart management and big-data analytics. Its applicability is demonstrated in the three lighthouse cities of the project, Nantes (France), Hamburg (Germany) and Helsinki (Finland).

## 1. Introduction

Cities are facing the population growth issue, which is becoming one of the main challenges for small, medium and large urban areas. According to what the United Nations stated in [1], it is expected that by 2050, more than 55% of the world’s population will live in cities. This translates directly into a 68% increase. Therefore, a new paradigm is necessary so that the Smart Cities and Communities concept can be applied, the two main reasons being the support of this growth from an environmental and sustainable perspective, and the assurance of the citizens’ quality of life.

According to the European Commission [2], “a smart city is a place where traditional networks and services are used in a more efficient way with the application of digital and telecommunication technologies for the benefit of its inhabitants and business”. In short, the creation of digital and intelligent services will be (or even nowadays is) essential for the efficient management of the cities’ resources. Among them, energy, water or transport could be mentioned. The implementation of Information and Communication Technologies (ICT) that support decision-making tools and services/verticals from a holistic point of view (e.g., smarter urban mobility, waste disposal and energy management considered together) is, hence, the mechanism to provide more efficient ways. However, still, there is a lack of interoperability and interactivity between verticals that need to be solved whenever creating more sustainable-friendly cities.

Under the digitalization paradigm, the implementation of Open Standardized Urban Data Platforms becomes crucial. Its definition is basically “the implemented realization of a logical architecture/design that brings together vertical data flows within and across city systems on a horizontal layer” [3,4]. The means to exploit the Urban Platform is via the integration of new technologies, such as Internet of Things (IoT), Machine-To-Machine (M2M), cloud, mobile or Big-Data, among others [5,6]. Thanks to them, the data interoperability between verticals (e.g., mobility, governance, energy) would enable the deployment of the aforementioned holistic digital services. As well, openness aspects are also crucial in order to provide data accesses. Therefore, a set of requirements need to be established [2]:Holistic focus: All these services should be offered within a multidisciplinary context and as-a-whole for a single access point.Quality of life: As stated before, one of the main objectives of Smart Cities is the citizens’ quality of life. Therefore, the Urban Data Platform should provide digital services that address this topic via integration of technologies (above of all, taking into consideration the digital era).Efficiency: Defined as the capability of managing local resources so as to improve the sustainability and reduce the costs.Interoperability: Data exchange and services integration are pivotal as explained before. The avoidance of vendor lock-in and the use of standards are main objectives.

From these requirements, it might be concluded that interoperability is the primary challenge. Multiple and heterogeneous vendors co-exist and even protocols are not homogenized among verticals (e.g., energy and mobility transmit data in different ways). Furthermore, cities are usually managing multiple platforms at the same time, e.g., transportation panels for real-time visualization of the fleets or dashboards for energy generation and distribution networks. Thus, there is still a gap in achieving a common consensus or standard dealing with interoperability. The importance of the interoperability can be viewed from the current works that are focused on this topic. For instance, in [7], the connection of co-existing IoT platforms under multiple domains is rendered, trying to integrate them by using software, from a systematic perspective. In contrast, others, like [8], are fostering the IoT interoperability via new open frameworks applicable at large-scale. Finally, studies such as the conducted in [9] are looking for ensuring interoperability from design, according to a similar conceptual approach.

In this sense, mySMARTLife (GA #731297) [10] is tackling the interoperability from a cross-domain design perspective. The Smart City and its digital agenda, as well as a holistic point of view, should be considered in the first stage. Then, by establishing an Open Specifications Framework [11], the project aims to demonstrate, in the Lighthouse Cities of Nantes (France), Hamburg (Germany) and Helsinki (Finland), that the openness and interoperability of Urban Data Platforms are ensured, integrating the heterogeneous domains and data sources. However, it is important to establish the framework definition. This is not a real realization of the urban platform, but a conceptual approach that provides the mechanisms to define the methods and procedures to implement the urban platform. It could be confused with commercial products, but the framework is a step back, trying to tackle the lock-in issues that appear when using commercial developments that are not coupled with the cities requirements. Thus, the definition of the framework is focused on three main pillars:Implementation of Open Data and Open standardized APIs. (application programming interface)Capabilities for Interoperability & Data integration under standardized data models.Privacy and security.

Besides, this Open Specifications Framework provides not only standard interoperable mechanisms, but also replicability and scalability features. Here, it should be noted that data are essential for the decision-making process. Therefore, the challenge is how to deal with the variety of data in a unified way and into a common operational framework.

In order to achieve the aforementioned features, a five-step methodology has been followed. The first step has been the identification of existing initiatives at European level (including on-going projects), as well as research results, concluding in the background that is described in Section 2. Once strengths and weaknesses had been identified, the requirements of the cities (step 2) were collected with the aim of following a bottom-up approach. That is to say, from the cities’ necessities to the design of the framework, instead of top-down, where unfeasible concepts could provoke lock-in in terms of capability to be implemented. Nevertheless, at the same time, being broad enough to foster replicability and scalability where specific solutions fail. Thus, the mySMARTLife Open Specifications Framework is defined (stage 3) considering both the existing initiatives and the cities requirements, as illustrated in Section 3. mySMARTLife contributes in terms of providing open standardized APIs and data based on global standards, where third party developers may integrate new developments, other cities may adopt the framework and data are “self-discovered”. As well, the data management mechanisms are clearly defined, identifying the urban platform boundaries. However, the Open Specifications Framework needs to be validated (stage 4). Hence, the three lighthouse cities of Nantes, Hamburg and Helsinki have implemented their own reference architecture based on the framework, as detailed in Section 4. This fact demonstrates the flexibility of the results to different cities, multiple architecture schemes and current platforms to be extended and/or new developments. Additionally, the capability for interoperability is demonstrated and validated where services might operate from platform-to-platform as long as a minimum set of requirements is complied with. As last step and after the demonstration campaign, the lessons learnt were extracted so as to determine not only the benefits, but also the limitations that the mySMARTLife Open Specifications Framework usage provide, described in Section 5 of this article.

## 2. Background for Smart City Urban Platforms

### 2.1. Existing Initiatives at European Level

Urban Data Platforms and the interoperability aspects are not neglected from the European Commission and other initiatives. In this way, several analysis, norms and frameworks are available. Three of them are included here: European Innovation Partnership–Smart Cities & Communities (EIP-SCC) [3], International Telecommunication Union (ITU-T) [12,13] and ESPRESSO [14].

#### 2.1.1. European Innovation Program—Smart Cities and Communities

Starting with the EIP-SCC [3] initiative, it is interesting to mention that it is a work group supported by the European Commission focused on Smart Cities. Through two of its work streams (City Needs and Industry Partners), it defines some design principles and initial reference architecture for data platforms at a city scale. One of the main goals is to enhance standard architectures to assure interoperability. Under this purpose, the EIP-SCC defines a “capabilities” architecture [3,11], whose description in detail is not an objective of this paper. It establishes eight horizontal plus two vertical layers. This paper does not pretend to describe the details of the capabilities but, in short, the horizontal core is the following:(1)Data exchange and ingestion, as well as field level communication establishment.(2)Support the devices data exchange by assets management.(3)Storage of data, aggregation and calculation of analytics.(4)Orchestration of processes and services, allowing human-machine interaction.(5)Generic City & Community capabilities, for general purpose services.(6)Specific City & Community capabilities, for specific verticals, e.g., mobility.(7)Focused on visualization of information for decision-making.

Thanks to these capabilities, a set of specifications are overcome [2,3]. There is no direct link between the capabilities and the specifications, but the realization of these capabilities deals with the requirements that are listed below:-Interoperability as main pillar for urban data and infrastructures.-Replicability.-Scalability, without creating technical and cost challenges.-Open standardized APIs and SDKs.-Capability of Real-Time data processing.-Integrability of functional and technical requirements.-Finally, in order to overcome the aforementioned requirements, field equipment that measures the field conditions (IoT devices).

With this multi-layered framework proposal, mySMARTLife improves it by including, on one hand, privacy and security mechanism before aggregation, just at the data exchange and ingestion level where it needs to be applied. On the other hand, northbound interoperability is not clearly covered by the EIP-SCC initiative. For this end, standard data models and linked data are required, which is also established within the mySMARTLife Open Specifications Framework.

#### 2.1.2. International Norm ITU-T UNE 17804:2015

Secondly, the international regulation ITU-T UNE 17804:2015 [12] is defined by the International Telecommunication Union within its Telecommunication Standardization Sector, which is in charge of the coordination of the standards for telecommunications. This norm sets reference architecture under the concept of a holistic view.

As displayed in Figure 1 [12,13], it is composed of six main layers [11], namely, from bottom to top:-Collection Systems level addresses the physical layer as field devices in charge of data monitoring, e.g., wattmeter. Besides, other data sources are also included, such as social networks, participant portals, etc., in conclusion, raw data.-Acquisition/Interconnection layer, with the aim of integrating raw data from previous layer and thus ingesting to make data available for upper levels, having in mind semantics.-Knowledge layer whose function lies in the data management.-Interoperability layer, being key level for the provisioning of open APIs to allow services deployment, as well as other data consumers.-Intelligent Services, representing the verticals of the Urban Data Platform.-There is an additional transversal Support layer, dedicated to configuration and maintenance work.

#### 2.1.3. ESPRESSO Project

Last but not least, the ESPRESSO project [14] (related to the EIP-SCC Work Stream 3) aims to standardize and collect lessons learnt from finished projects. Among them, standards, conclusions and/or best practices are the most common ones. In this sense, by combining these outcomes and conclusions, it defines a “conceptual standard framework” for Smart City Urban Platforms, such as the one illustrated in Figure 2 [15].

As observed, the concept is similar to the aforementioned approaches. The main differences lie in the sensing layer (combination of physical devices and communication mechanisms). Similar to ITU-T, but contrary to EIP-SCC, the IT Enabled Services layer is merged, while EIP-SCC distinguishes among multiple layers for the high-level services (including Human-Machine Interaction), with the exception of the visualization, which is integrated into a Business Layer.

#### 2.1.4. Other European Projects

Not only are the aforementioned initiatives working on a common definition of Smart Cities Urban Platforms, but also, there exist several projects trying to address the conceptual approach, as well as interoperability matters. Some examples are included within the Synchronicity initiative of the European Commission [16,17] under a common goal to ensure interoperable digital data platforms. Some of them are SmartSantander [18] or Organicity [19]. Concepts for Smart Cities are commonly developed, integrating technologies like IoT, Big-Data, and new concepts like Web APIs and smart objects. The answers keep focus on industrial purposes in order to encourage the technology industry to sit together to reach a consensus for interoperable solutions.

SynchroniCity [16] mainly aims to the creation of digital solutions where cities and businesses may work together to develop IoT- and AI-enabled services to improve the lives of citizens and to grow local economies. For that end, interoperability aspects are significant to reach vendor-neutral and technology-agnostic solutions. It looks for several interoperability features, such as Context Information Management API or Shared Data Models. It also provides reference architecture based on the previous ones [17], where interoperability is managed in terms of northbound and southbound interface (services and data ingestion levels, respectively).

SmartSantander [18] also bets on interoperability through interconnecting infrastructures and gateways able to exchange data between different actors, within an IoT-based architecture. It collects 3,000 heterogeneous observations per day. Besides, Organicity [19] looks for similar solutions for data integration by including data interfaces.

However, all of them fail in the integration of multiple domains into the Smart City platform. It is true they already integrate multiple domains, but they miss others. For instance, SmartSantander [18] includes transportation and social (tourism information), while energy needs to be improved [20]. Indeed, according to [20], current architectures are not able to efficiently face all the challenges that a Smart City presents.

### 2.2. Scientific Background

The various initiatives and projects are not only an important source of knowledge and development in terms of urban platforms. There are also scientific research projects trying to answer the necessities of current Smart Cities in the form of urban platforms. However, there is still not a consensus on the lack of a holistic view of the urban platform. For instance, the work in [21] is focused on providing a platform for noise monitoring. Something similar is happening for lighting, such as in [22]. Even a third pillar, which is important in Smart Cities, as it is the waste management [23]. None of them integrates the diverse pillars of the Smart City.

Nevertheless, the new paradigm within Smart Cities requires a global picture of the city and determines crossed effects among verticals. For instance, noise levels are partially dependent on the mobility status of the city. Waste management also needs the mobility pillar to calculate the truck routes. Hence, other studies have been focused on this global overview. Smart City taxonomy is formed by multiple dimensions, like buildings, city planning, etc., as detailed in [24]. Then, Smart City is viewed as an IoT network where all the elements are connected through IoT sensors [24], although it misses the common data representation as IoT protocols are heterogeneous. As an example illustrated in [25], a real-time “sense-able” city is defined. Indeed, one of its main conclusions is the requirement for semantic and linked data [25]. However, it mainly establishes a semantic Web with no standard model for other kind of data (e.g., non-IoT data). Besides, it also presents an architecture for “live Singapore”, but it lacks the value and knowledge extraction, being focused on semantic filtering to process data.

Having said that, there exist other studies working on this paradigm. The authors in [26] present a novel web-based platform to that end, which includes data from heterogeneous sources via semantic integration. However, it is focused on public data and social media, which still is not the single objective of a Smart City, and is missing data from private companies (e.g., energy generation plants). Something similar is considered in [27] with a Resource Description Framework (RDF)-based language for data vocabularies and relational data. As the authors already state, selecting the most suitable ontology is not easy and emerging technologies are important, such as SensorThingsAPI in the case of mySMARTLife project.

Additionally, other scientific publications with focus on interoperable solutions are also analyzed. For instance, the authors in [28] already consider the multi-domain in terms of sensors and IoT equipment for data collection in Smart Cities, highlighting the issues of data integration. To overcome it, a Web Service named InterSensor Service is presented. The main objective is to connect heterogeneous data samples via OGC (Open Geospatial Consortium) specifications [29], making use of the observations (similar to the proposed mySMARTLife framework) to integrate explicit semantics in data interoperability. Nevertheless, despite providing an interoperability solution of Smart City Data Platforms, it fails in the proposition of a common framework under which the services, dashboards, analytical tools and applications could work in the same harmonized way, increasing the interchangeability. Mainly, the focus is on the connections to the field level and encoding the data samples “on-the-fly” through standardized data models to provide unified data to the higher levels in the architecture. Moreover, the InterSensor Web Service aims at platform-to-platform interoperability [28] by the inclusion of adapters, while mySMARTLife goes a step forward considering interoperability at multiple levels to ensure sensor data integration, semantics and also interoperable services.

Another example that goes in the same direction than before is the work presented in [30]. As it is remarked, data harmonization via standardized data models is pivotal. The main conclusion is the necessity of considering ontologies as design pattern, which is one of the design principles for the mySMARTLife Open Specifications Framework. In this sense, based on standard data models, it provides an ontology by combining Smart Cities domains [30], which results in an integrated Ontology Design Pattern (ODP) relatively complex. A particular implementation for heterogeneous mobility data is depicted in [31], where Big-Data analytics are enabled by the data integration.

Continuing with data integration research results, authors of [32] analyses an approach for spatiotemporal data representation integrated into a reference architecture for event-driven applications. Again, heterogeneity and multi-domain data are particularly challenging in order to provide unified decision-making tools. In this sense, geographic information is important, being not fully addressed in current solutions for spatial data analysis. However, the results from [32] show services that are triggered by a geographic event (e.g., traffic accident), while a city requires different levels of analytics: real-time processing without the need of a trigger, event, schedule…, which is considered within the proposed framework in this paper. In fact, according to the comparison performed within [32] for multiple smart city platforms [33,34,35,36,37]; scalability, privacy and security, processing at different levels (batch, real-time) and analytics are features that are still not fully addressed, then failing in the holistic view of the Urban Platform as Smart City solution. It establishes SensorThingsAPI as a more efficient communication protocol than the previous ones. Finally, RASCA architecture [32] is proposed where some aspects are neglected such as privacy, security or northbound interoperability, among others.

Last but not least, Ref. [38] relies on a new data management procedure where combination of data promotes application innovation. In contrast to previous researches, this specially treats privacy and security, which is pivotal, even enhanced with the General Data Protection Regulation (GDPR) [39]. In the same line, Ref. [40] describes an ISO37120 standardized architecture for urban platforms where anonymization is already included as one of the key aspects of these solutions.

### 2.3. Main Conclusions from the Previous Framework Analysis

From the previous analysis of initiatives, norms, projects and researches, several conclusions may be extracted from what a Smart City Urban Platform should comprise [11]. Summarizing:Monitoring, via sensing devices, is necessary in order to get raw data from equipment deployed throughout the city. Multiple and heterogeneous data take place here, including various and diverse city pillars, such as energy or mobility. Moreover, sources, protocols and data models diverge among them (e.g., IoT devices transmitting in LoRa (long range) or 3D city information based on the CityGML standard).According to the raw data heterogeneity, data ingestion becomes crucial in order to integrate all the incoming information. In this sense, interoperability at field level is an indispensable requirement, which depends upon the data drivers and communication interfaces.Data management is also a common aspect to be considered. Homogenization of the gathered information via standard data-models, calculation of added-value information via indicators and advanced analytics, relational and non-relational storage and privacy/security aspects are essential at this stage.Smart City solutions differ from industrial solutions in their requirement for linking data with spatial context. Practically all the data that cities manage is related to a specific location. Spatial data are not only the location of the sensor but also its coverage area, shape and so on.Not only field level interoperability, but also at data consumers (northbound) level is necessary. For that end, the implementation of open APIs, open Data and open SDKs should be considered in any Urban Data Platform implementation. Also, compliance with the INSPIRE requirements sets the requirement to be compliant with ISO/TC211-standards.Finally, intelligent services with a main focus on the end-users in order to support participation, transparency and trust among citizens’ help the urban planners with their job.

From these bullets, the biggest contribution of mySMARTLife is established to be through interoperability, which is a key aspect for most IoT systems. Data integration (communication level), data management, data sharing (open data and open APIs) and services deployment are essential for complying with city/ citizen-specific requirements.

#### mySMARTLife beyond the Current Practices

Most of the existing smarty city platforms are focusing on IoT only or semantic Web. Thus, these are not “real” Urban Data Platforms. Urban Data Platforms handle also static and metadata, while the latter is neglected most of the time anyway. IoT platforms will help city administrative organizations mostly in a single domain for their specific task. Their most important thing would be that these IoT platforms provide open standardized APIs to connect easily to the urban data platforms forming a system of systems. Since all countries in Europe have to fulfil the INSPIRE initiative, a good starting point for Urban Data Platforms is the spatial data infrastructures which already exists. Having already long-time experience in data management, using open standardized APIs, including metadata handling these spatial data infrastructures, can easily be extended for real time data forming an open standardized urban data platform.

Under this approach, mySMARTLife Open Standardized Urban Data Platform aims to define an interoperable framework to be considered from a design perspective in order to ensure data integration, exchange, INSPIRE directive compliance and a holistic view. Furthermore, replicability and adaptability are essential. This means that cities with digital platform can easily adapt their current deployments with the Open Specifications Framework as demonstrated in the three Lighthouse cities of the project. To overcome the aforementioned problems, interoperability is established at three levels: southbound, northbound and semantic; that is to say, raw data integration, services data sharing and homogenization of data samples.

Furthermore, the mySMARTLife Open Specifications Framework also re-defines the above explained initiatives or standards to simplify complex concepts and/or bridge the gap of some others. mySMARTLife considers the Smart City as a holistic concept where interoperability should be covered from the sensors to the multi-domain services, passing through semantics and enrichment of information in contrast to the InterSensor Service that considers the data drivers as an interoperable solution [20]. In this sense, it also tries to decrease the complexity of the Smart City itself by reducing the number of “isolated” platforms that an urban planner would need to operate. Instead of having individual solutions for each vertical, such as [22,23], mySMARTLife framework fosters the correlation between verticals to obtain a more real picture of the city to make better decisions (e.g., effects of the pollution due to mobility).

Moreover, mySMARTLife establishes clearer reference layers in order to help developers and integrators at the time of implementing open and interoperable Smart City Urban Data Platforms. In this sense, it avoids complex data models like the one in [30], complex conceptual approaches such as the EIP-SCC [2] or simple interpretations of urban platforms. Therefore, the major step forward is the definition of a full interoperable framework that includes sensor integration, capability of services exchange platform-to-platform and harmonization of information. Finally, it also accounts for privacy, security and data quality aspects (as well GDPR [39] issues).

Having this holistic view in mind, the proposed framework supports developers and integrations at the time of establishing a common conceptual approach where not only interoperability is included, but also openness, data sharing and analytics for decision-support. For that purpose, standard data management procedures are also defined to guide involved stakeholders. Most of the solutions fail on the data management mechanisms, not providing standardized ways to treat the information, despite harmonizing data samples. This affects replicability and scalability of urban platforms; concepts that are fundamental criteria from design.

Last but not least, it should be mentioned the multi-domain approach of mySMARTLife. While within Industry 4.0 the projects are dealing with IoT platforms for industrial interoperability, mySMARTLife already considers cross-domains to create an open ecosystem where multiple stakeholders are possible. Thus, Industry 4.0 solutions could be considered as not applicable in Smart City projects, due to their specific character.

## 3. mySMARTLife Open Specifications Framework

Taking into consideration the aforementioned initiatives, the existing and current gaps, mySMARTLife has redefined the reference architectures to establish an Open Specifications Framework. Note that this framework is not an architecture, but a conceptual approach under which the reference architectures may be designed from city to city. This means the mySMARTLife framework extends replicability, being flexible in comparison with reference architectures that must be implemented. Flexibility is defined in terms of capability of adaptation to the city needs, providing a full context where Smart City requirements may be covered.

Figure 3 [11,41] describes the proposed Open Specifications Framework. This framework is built upon the previous initiatives while, at the same time, redefining some of the concepts to present the Smart City Urban Platforms with a broader overview.

The framework, layer by layer, is described as follows:The sensing layer is similar to that of previous approaches, referring to the physical environment where metering equipment (IoT devices) and connectivity are included.The drivers’ layer is proposed in contrast to the previous frameworks. Although the aforementioned initiatives keep in mind the drivers needed to communicate with the field level, mySMARTLife framework explicitly refers to this layer in order to ensure the data acquisition. Then, it enhances the southbound interoperability by means of data integration procedures based on open and standard communication protocols. Therefore, it is not only a matter of data collection, but also harmonization of data samples before being ingested into the data repositories. It also broadens the aforementioned definitions with data buffering mechanisms to avoid data losses in the data ingestion processes. Therefore, intermediate and temporal timeseries are buffered to allow data recovery.The surveillance layer is completely new. It deals with the GDPR concerns (privacy and security) by allowing for integration of anonymization, aggregation or any other technique that ensures the GDPR compliance. Moreover, the surveillance concept helps to increase data quality through quality checks before storage (e.g., out of range values). In this sense, secure, private and accurate data are stored in persistent repositories, ensuring the quality that is sometimes neglected.The data layer that remains as a data storage and analytics layer, but increasing the level of storage not only to dynamic samples, but also repositories like Geographical Information System (GIS), static information from the city and other repositories that the city could require. Moreover, the corresponding Extraction, Transformation and Loading procedures (ETLs) are included with the objective of managing these repositories. Finally, analytics aggregated data in order to extract intermediate results (such as indicators) that could be useful for upper layers.The interoperability layer, whose concept is re-used from ITU-T for northbound interoperability in terms of open data and APIs. However, it also includes an open Software Development Kit (SDK) for third party developers and the myData concept, which deals with GDPR as well. In this sense, myData provides the mechanisms to explicitly and affirmatively handle the consent to the use of personal data and take advantage of the usage of high-level services.The intelligent services layer, being the vertical domains where the high-level services apply within the Smart City. From energy management to e-governance, mobility to citizen-focused services, this layer is where the interaction between city stakeholders is created.

### 3.1. Data Management Procedures for Interoperability

As described above, mySMARTLife framework describes new concepts for Smart City Urban Platforms, highlighting the interoperability mechanisms that a Smart City should fulfil. For this purpose, to complement the framework a data management procedure is also determined where the different stages and interoperability levels are highlighted. Figure 4 [11,41] displays how the framework manages data from the collection to the consumption.

Basically, the IoT sensors and other field measurement equipment publish data that need to be collected by means of data drivers. They import and integrate the information (drivers’ layer) to cover the first level of interoperability (southbound). Data require to be modified according to GRDP constraints and/or quality checks (surveillance layer) in order to store and calculate the analytics (data layer). Furthermore, next step exposes data [41] to overcome with the next level of interoperability (northbound in the interoperability layer) by the publication of the datasets. Finally, services and third parties consume data to provide intelligent services and/or applications (intelligent services layer).

In conclusion, mySMARTLife deals with the three levels of interoperability described below:Foundational interoperability to assure data exchange between two information technologies, without the capability to interpret the data.Structural interoperability that defines the structure or format for data exchange (i.e., the message format standards). Uniform flow and meaning of data are preserved and unaltered. Structural interoperability defines the syntax and ensures interpretation at the data field level.Semantic interoperability provides interoperability at the highest level, which is the ability of two or more systems to understand data thanks to common data models. Semantic interoperability takes advantage of both the structuring of the data exchange and the codification of the data including vocabulary so that the receiving data systems can interpret the data in an intended way.

#### 3.1.1. Foundational and Structural Interoperability

Foundational and structural interoperability within mySMARTLife are ensured in terms of southbound and northbound interfaces. In the first case, the provisioning of data communication drivers to connect field equipment is the main challenge. As stated above, multiple and heterogeneous vendors are currently in the market. Nevertheless, the predominant technology in Smart Cities nowadays is the IoT. Therefore, southbound interoperability is preserved by the availability of drivers able to communicate in the most used protocols, for instance LoRa.

In terms of northbound interfaces, mySMARTLife relies on common data interpreters in order to provide open APIs that enable data consumption from services. In this sense, open and standard APIs, as well as open data based on CKAN as a data index or catalogue extend its use to provide metadata of the APIs to provide a self-understandable interface for third parties when integrating new services.

#### 3.1.2. Semantic Interoperability via SensorThingsAPI Semantic Model

Up to now, interoperability has been established on two levels: northbound and southbound. Nevertheless, this is not enough for data processing and interoperability from bottom to top. Semantic interoperability is the third level and, perhaps, the most important one. The reason is the capability to interpret and exchange data (e.g., with other urban data platforms or third-party services) in a common vocabulary or ontology [42].

mySMARTLife also defines this interoperability level within the framework, which is embedded from the drivers layer to the interoperability layer. In this way, SensorThingsAPI [29] has been selected among other data models, whose entity-relationship diagram is depicted in Figure 5 [29]. It is an Open Geospatial Consortium (OGC) standard and, nowadays, it is also standardized by the ITU-T [43]. Among the set of available technologies, such as the work on [30], its selection is characterized because it is open, standard, has the capability to interpret geospatial data, unifies data samples into observations, is able to cover multiple domains and complies with w3 standards.

SensorThingsAPI standardizes the data observations [29], as well as metadata in heterogeneous and diverse data ecosystems. This is pivotal for the mySMARTLife framework considering the heterogeneity of the data sources (3D city models, energy metering, CANbus e-car sensors, charging stations and many others), as well as the verticals (Smart City pillars).

The SensorThingsAPI model is powerful: it provides a single data model to integrate all kinds of observations. At the field layer, either a piece of data is integrated directly or using connectors to fetch the data from IoT devices or from IoT supervisors, calculating KPIs (Key Performance Indicator)*,* or integrating KPIs from those supervisors’ reporting. For the monitoring of the project, most of the KPIs are calculated by the supervisors and integrated with specific procedures because existing supervisors does not always provide APIs to connect to.

Nevertheless, it lacks the provision for a common dictionary to represent data. All observations in SensorThingsAPI have the same fields/structure, but there is no way to objectively know what they represent because this information is defined as “ObservedProperty” (cf. name and description fields). For instance, an ObservedProperty name could be “TEMPERATUR”, “LÄMPÖTILA” or “TEMPÉRATURE” in German, Finish and French, meaning the same, but with a different representation. Then, the mySMARTLife Open Specifications has also harmonized this subjective specification into an objective one where all the data samples would be characterized with the same metadata (see next section) to comprise incoming information.

According to SensorThingsAPI nomenclature [29,43], an **Observation** (for example: an indoor temperature of 22°C) concerns a Feature of Interest (a room in a dwelling or a pedestrian cross in mobility). A **Datastream** (daily temperature curve) groups several Observations provided under the same **ObservedProperty** (temperature) and detected by the same **Sensor** (thermometer inside the room). The **Datastream** relates to a **Thing** (temperature controller system in the dwelling case or a traffic camera for the pedestrian cross example), which can be linked to a **Location** (room location) and, in the future of SensorThingsAPI, to Tasks (control commands) [11].

Finally, SensorThingsAPI makes cities complaint with the INSPIRE Directive by extending INSPIRE to the IoT [44]. The European Union INSPIRE Directive laid down Spatial Data SonsorThingAPI is able to overcome these specifications, and this, benefits the implementation of Urban Data Platforms to be interoperable, standard and INSPIRE-compliant.

#### 3.1.3. Data Sharing: Metadata for Structural Interoperability

Semantic interoperability is not only a matter of standard data models (or representation), but also data interpretation [42]. For this purpose, metadata should be also considered when sharing, i.e., northbound interoperability based on open APIs. Metadata provide mechanisms to auto-interpret and self-discover data [41]. Urban platforms are expected to support heterogeneous and multi-domain sensors, a fact that complicates the data understanding, hence requiring dynamic data models that are being adapted along the urban platform life cycle (i.e., representation of current and future data types, instead of fixed data models). For that end, data enrichment paves the way to extend current incoming data streams with additional properties to conform linked data, contextualizing these data.

Therefore, although SensorThingsAPI is the basis to data representation, other taxonomies close to the domains of a Smart City are analyzed. In this way, third party data consumers, such as services or developers, require data interpretation mechanisms. That is the reason why, within the interoperability layer, for providing open APIs and SDKs, JSON-LD (JSON Linked Data) [44] is also considered to add the capability of self-discovering data.

The solution adopted within mySMARTLife is to provide a link to the JSON-LD definition within the JSON objects [45]. The advantages of the solution are being simple and HTTP/MQTT (Hyper Text Transfer Protocol/Message Queuing Telemetry Transport) compliant, but at the expense of being a little intrusive. A web server offers the service for the JSON-LD files that define the metadata for objective representation of data samples (e.g., temperature common naming stated above) and, thus, deploy a self-explanatory API for third parties. Then, the operation connects the web server when a data sample is received to enrich and harmonize its representation under a common and already established vocabulary. The solution is selected from other two possibilities that were discarded, on the one hand, for being very intrusive with high level of redundancy and, on the other hand, for not being applicable for MQTT transport.

## 4. Validation and Demonstration in Smart Cities

As it has been remarked, two interesting features of the framework are the replicability and the flexibility of the framework. For that end, the framework has been implemented in the three lighthouse cities of the mySMARTLife project: Nantes, Hamburg and Helsinki [10]. As explained previously, the framework is a conceptual approach that should be realized in specific reference architectures. Moreover, it should be noted here the adaptability benefit that was mentioned before. Cities that already operate with urban platforms are capable of adaptation when necessary. Next sections map the individual architectures into the Open Specifications Framework for each city [11].

### 4.1. Nantes

Starting with Nantes, the main purpose is to extend the current architecture with new features, according to the necessities of the city. They are focused on the integration of new data related to the three main pillars of the mySMARTLife project [10] (energy, mobility and ICTs), as well as deployment of new added-value services, such as smart data on mobility or energy data lab initiative, among others.

In order to overcome these needs, the urban platform architecture is defined as Figure 6 [11], being perfectly mapped into the framework. Table 1 includes the layers and their functionality, as well as the mapping into the proposed framework. Then, Nantes Urban Platform extension is completely compliant with the framework with the aim of managing the heterogeneous data samples and providing added-value services as explained below.

The core of Nantes’ mySMARTLife Urban platform extension is based on an OGC-defined model compliant with implementations of SensorThingsAPI standard (interoperability layer). KPI calculation and data analysis are performed on such core model (business layer). Data are exposed for both incoming and outgoing streams through APIs administered by API managers (security layer). Dashboards (IT services) are built on top of the core model (internal dashboards with performance aspects) and open APIs (third-party dashboards). Accesses to the dashboards are regulated by identity and access managers (security layer).

Nantes’ Urban Platform also integrates data through connectors, into the single data model of SensorThingsAPI, which provides a uniform access to observation data. Both observation and referential data are integrated. At the core of the platform lie in two distinct data sets: one is directly completed by field data, the other is dedicated to exposition and use by the end-users or third-party applications; only data eligible to being exposed are copied from the former to the latter. While private data is not a core aspect of Nantes’ Urban Platform, still, this mechanism allows controlling data privacy as per Nantes Metropole’s role regarding rules and regulations: sensitive data can be collected and used for non-sensitive KPI calculation, then, only the KPIs would be transferred to the exposed data set and accessible as open data.

Regarding access control, when data are fetched by connectors on the Urban Platform side, only expected data are collected and access control is performed. When data are pushed to the urban platform, mechanisms aside SensorThingsAPI must be enforced: the SensorThingsAPI model does not cover security aspects. For instance, different data publishers could publish directly into the SensorThingsAPI model, overriding the notion of domain. Within mySMARTLife, we did not experience this issue because, on one hand, all data publishers are integrated through connectors which provide this extra layer of security, furthermore reinforced by the API manager—which handles the user access rights through API keys, and the HTTPS communication protocol with private certificates; on the other hand, accesses to the SensorThingsAPI are managed through API managers, which allow restricting access to publishing and reading APIs on a per-user/token basis.

Finally, the Open Specifications Framework has several layers, where different stakeholders can contribute; it is very important to settle cross liability questions. In this project, Nantes Metropole and Engie signed a bilateral agreement on top of the project’s grant agreement to settle specific points of the intellectual property for data and developments.

### 4.2. Hamburg

Following a similar approach like Nantes, Hamburg is currently working with an Urban Platform in charge of data storing and analyzing unit for open and non-open data from different authorities, third parties and few sensor data. Furthermore, one of the main datasets is related to geospatial information classified into several categories [11].

This approach has been extended following the open conceptual framework defined in mySMARTLife Project into System of Systems approach/architecture; extending an OGC-based Urban Data Platform with a similar one, based on the global oneM2M standard, as illustrated in Figure 7 [11]. This standard also allows heterogeneous systems to be connected.

While cities have an interest in applications and standards with, i.e., geospatial backgrounds like OGC, many industry players need a standard like oneM2M providing additional features like device or access management. Since both standards have their strengths, it is worth to build an integration or interworking. While OGC and its SensorThingsAPI (STA) is easy to use and provides excellent semantic description through its data model, oneM2M provides access control and data routing.

Both systems are running on a modern micro-services architecture and comply with the idea of an Open Standardized Urban Platform where each have their advantages, and the combination uses the best of these two technologies. The OGC is strong offering an almost complete set of standards for different purposes for an open standardized urban platform. These standards apply i.e., for metadata (CSW), static data sets (WMS—Web Map Service, WFS-T—Web Feature Service, WCS—Web Coverages), web processing services (WPS) and since recently also for sensor data management including the spatial relation (SensorThingsAPI). However, the standard does not cover the very “southern” part in an IoT world, where device provisioning and IoT harmonization are required. Here, the oneM2M standard focuses on the technical harmonization and orchestration of different domains, including aspects like device management or authentication. This oneM2M framework is based on open standardized APIs. The central logic of oneM2M uses a Common Service Entity (CSE), which is acting as central orchestration instance. The CSE receives data from different domains or sensors via Application Entities (AE) or Interworking Proxies. These data can be accessed from applications or they are distributed to interested parties by the CSE through sophisticated publish/subscribe mechanisms. OneM2M also provides integration with different authentication mechanisms and has an own access policy mechanism and description. Through this way it is possible to describe fine-grained authorization schemes for sensor data access. OneM2M provides an extensive framework for gathering, routing and orchestrating data. Beside data-routing, oneM2M covers also areas like network control functions and device management. There are also initiatives to define so called device classes. These are data structures that define domain specific data semantics. Nevertheless, this work is in a rather early stage of development and currently there is no alignment with so SI or ISO standards.

The innovative challenge is to combine both the OGC-based and oneM2M systems towards an Open Standard-based Urban Platform (OSUP) in order to provide the requested interoperable and future proof solution (see Figure 7). Both systems combined provide the requested open APIs and standards from the IoT device level up to the application layer and can deliver open data based on the common ontology. Besides the interconnection of the two systems, it requires the development of a common data exchange logic and data model.

The combination of the two standards provides advantages of a smart city ecosystem. Integration of hardware or service providers, e.g., from the IoT space that already use oneM2M or from the OGC world which is quick and simple. These partners have the advantage that there is no need to adapt respectively to the other standard. The OSUP provider on the other hand can select the more appropriate standard, based on service or usage requirements, to connect a service or hardware partner. Data sets requested from the system can be provided as “open” data in the data format of oneM2M or OGC. In case of certain advanced requests or enriched data more advanced OGC data model will be in place. This could be of importance for developers or certain industry partners.

The core of the data management of the Hamburg Urban Platform is divided into five modules: Data Web Services, Metadata Web Services, Processing Web Services, Data Analytics and Sensor Web Services. These are implemented in a multi-layer architecture as depicted in Table 2. Similar to Nantes, the Hamburg Urban Platform follows the specifications of the framework, which is applicable into a completely different architecture.

### 4.3. Helsinki

Last but not least, Helsinki, which is also part of the SynchroniCity project [16], has adapted the reference architecture to the framework. Figure 8 and Table 3 [11] show the reference architecture and its mapping into the framework, respectively. Like Nantes and Hamburg, Helsinki has extended an existing Urban Platform with the mySMARTLife new features following the Open Specifications Framework. Helsinki has developed the platform under the concept of City-As-a-Platform so as to link together several data-related services, providing a single point of service for developers, city officials and citizens requiring information in a machine-readable format. For the data ingestion, in terms of building energy, information requirements are derived from the Project Haystack terminology lists together with the Unified Code for Units of Measure (UCUM), which is based on ISO 18000 standard. On the other hand, electromobility interface protocols rely on the Open Charger Point Protocol (OCPP) and the ISO/IEC 15118 standard. After its collection, a formatting process is applied in order to be compliant with SensorThingsAPI, while CityGML v2.0 data model is also used to extend SensorThingsAPI in terms of 3D representation.

One of the major characteristics within the Helsinki urban platform is the MyData layer, which is partly the implementation of the “surveillance layer” of the framework. Basically, the citizen is put in control in all the data related to him/her. This ensures the authentication when provisioning sensor data to the platform. That is to say, data related to a personal owner is consented before being ingested in the urban platform.

Apart from the MyData tool compliance with the framework, the Helsinki urban platform is layered in five main levels as follows: City backend; Events, analysis & ETL; SmartCity API; Dashboard, city BI & apps; Authentication & MyData console. As stated before, Table 3 maps the aforementioned layers, including their functionality (mainly data integration, management, sharing, high-level services and configuration), with the framework layer that is applicable. Then, this is the third lighthouse city that demonstrates the flexibility and adaptability of the proposed framework to the specific urban platform requirements.

### 4.4. Interoperability Validation among Cities

From the previous sections, the replicability, scalability and adaptability of the framework are demonstrated. However, although interoperability is also included, it still lacks high-level services interoperability. Although its validation is still a future work to be overcome in the project, the assessment methodology has been established. Basically, it relies on the selection of use cases to be deployed across the urban platforms. The use case here means the realization of an added-value service that consumes data from the open APIs and, then, run the functionality. In this sense, six steps are defined for standardized interoperability:(1)Identify candidate standards and specifications based upon specific requirements;(2)Assess candidate standards and specifications using standardized, transparent, fair and non-discriminatory methods;(3)Implement the standards and specifications according to plans and practical guidelines;(4)Monitoring in compliance with the standards and specifications;(5)Managing change with appropriate procedures;(6)Documenting in open catalogues, using a standardized description.

According to these steps, two use cases are currently under development: (1) electricity consumption in public buildings (as data are openly accessible) and (2) street lighting (ownership by the municipalities), therefore avoiding privacy data issues. Then, based on the aforementioned SensorThingsAPI and linked data mechanisms, the interexchange of use cases is assessed through the “interoperability indicator”, ISO 37151:2015 [46]. Results are still not available, but, according to a Likert scale 1-5, the estimation is to reach, at least, 4, i.e., high level of interoperability.

## 5. Discussion

As it has been described along the paper, the Open Specifications Framework developed under mySMARTLife project provides several benefits that are explained below:Interoperability is the biggest challenge that is overcome, dealing with foundational, structural and semantic interoperability. The definition of the layered framework, standard data model based on SensorThingsAPI and linked data foster vendor-free integration in terms of raw data, data-sharing principles and integration of heterogeneous and diverse city verticals. Besides, it is based on holistic perspectives where the Smart City concepts are conceived for data integration, usability and sharing, covering the entire life cycle of the data. In this sense, the framework defines the data management procedures, data models and interoperable mechanisms from the sensors/IoT to the high-level services, passing through storage.Openness to ensure transparency and open data principles where entrepreneurship is also enlightened via open APIs/Data/SDKs. Self-discovering and dictionaries are included to ensure city-to-city or platform-to-platform data exchange capabilities.Replicability as being an Open Specifications Framework adaptable to current implementations, as well as setting up the basis for new developments. In this sense, follower cities of mySMARTLife demonstrate this aspect, such as Palencia, currently developing DigiPal project for digitalization of the city based on the same principles than the presented framework.Adaptability and Scalability as highlighted with the three lighthouse cities, where the integration of new data sources or use cases is a very simple task, when just making use of the connection interfaces.Ecological: The necessity for any type of data interpretation e.g., gateway equipment is omitted.

Secondly, it should be highlighted how this framework provides guidelines for other cities (like fellow cities) to implement their own Urban Data Platform. In this way, cities would only need to follow a set of steps:(1)Define the requirements according to the city needs, not only in current conditions, but thinking in a more scalable manner (i.e., future needs).(2)Map these requirements into the framework concepts about interoperability and openness.(3)Determine the necessary layers to comply with the specifications.(4)Design the architecture based on the aforementioned framework.(5)Select from the existing IoT platforms/systems the most suitable one that serves as a basis for the further developments (do not start from scratch).(6)Implement the interoperability and openness functionalities, as well as services.

Nevertheless, there exist some limitations:Sensor level interoperability requires the utilization of communication drivers, while the amount of existing protocols is huge; therefore, the number of available access points is limited. Then, the integration of new data must be adapted to the provided interfaces.Dictionaries are extensively defined for covering lots of data samples. However, within the mySMARTLife project, some pillars are not implemented; consequently new concepts could be necessary for linked data (i.e., JSON-LD).Services interchangeability relies on the availability of data. This depends upon the data acquisition system for each city; therefore, high-level services interoperability requires the compliance with the data sharing specifications.

## 6. Conclusions

Cities are currently facing one of the major challenges in decades, as is the population year-by-year growth, being estimated that 68% of the population will live in cities in the near future. This makes the use of resources in a more efficient way to ensure more comfortable, sustainable and well-being metropolis (but also in Smart Communities) essential. One of the mechanisms to do so is the digitalization and deployment of new technologies, which might be summarized in (data) integration and new services in a multi-domain approach. In short, digital transformation of Cities and Communities should be part of the digital agendas of Smart Cities and Communities.

This digitalization approach yields results in Open Standardized Urban Platforms as the final implementation. The ultimate objective is providing services to all groups of stakeholders, especially citizens and urban planners. However, to offer services from and across different domains, data in various blends like raw, integrated or aggregated are essential. Therefore, the interoperability aspects are one of the major challenges to be solved, which is the major contribution of the presented Open Specifications Framework. In this sense, a three-level interoperability approach based on foundational, structural and semantic concepts is encouraged to ensure data exchange capabilities at northbound, southbound and semantic levels. Thanks to this, new services are possible with the aim of fostering more livable cities.

Interoperability is sustained in openness as pivotal topic to be able to overcome a new transparency paradigm and data-sharing aspects. Nevertheless, it fails with privacy and security, as well as GDPR. Hence, data governance and management procedures need to be integrated as proposed within the in the surveillance layer of the framework, which is a very novel concept integrated within the mySMARTLife analysis.

Finally, its capabilities for scalability, flexibility and replicability are demonstrated and validated in the three mySMARTLife lighthouse cities (Nantes, Hamburg and Helsinki). Besides, follower cities, such as Palencia, are also taking the Open Specifications Framework as fundamentals for the digitalization processes. These great possibilities are conceivable thanks to the available freedom degrees. The framework provides a guideline for Urban Data Platform, being adaptable to the necessities and IT architectures of each city, as explained in Section 4.

As future work, after the final implementation and deployment of the urban platforms, as well as their services, interoperability will be tested platform-to-platform. That is to say, although the interoperability of the platforms themselves has already been validated, the interchange of services between the platforms is going to be also tested. Services from one city (or platform) will be used in the others to validate the semantic capabilities. These checks will lead to feedback the current design in order to extract lessons grasped, conclusions and propose improvements.

## Figures and Tables

**Figure 1 sensors-20-02402-f001:**
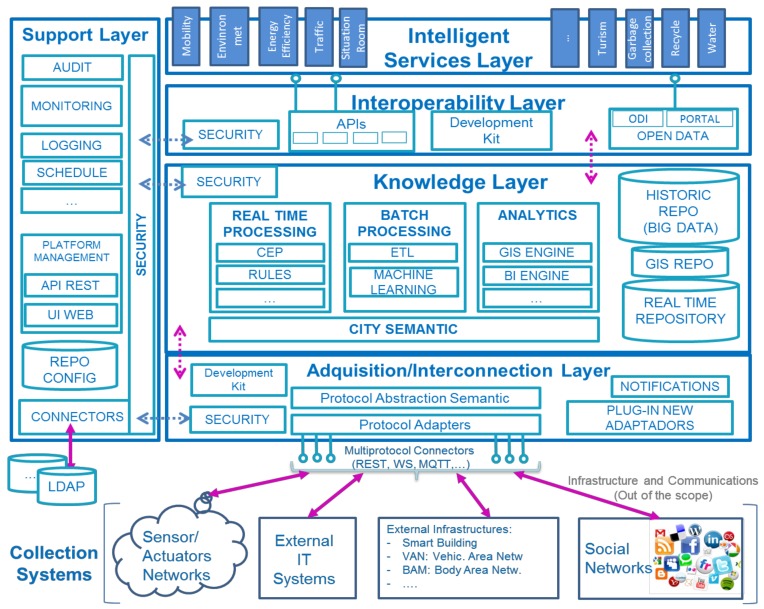
ITU-T (International Telecommunication Union) reference architecture for Smart City Urban Platforms.

**Figure 2 sensors-20-02402-f002:**
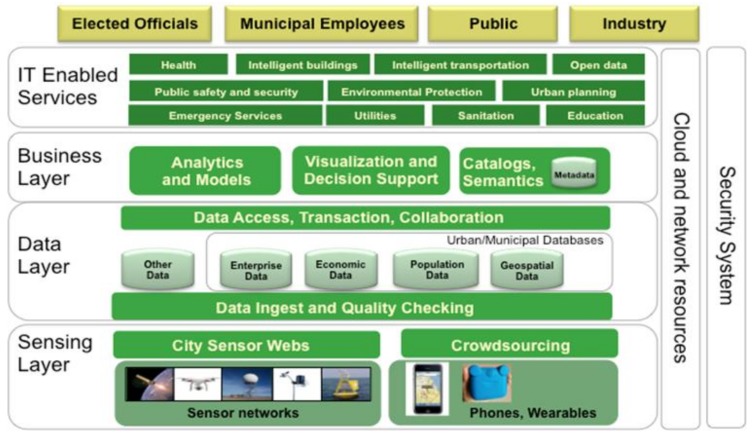
ESPRESSO project reference architecture.

**Figure 3 sensors-20-02402-f003:**
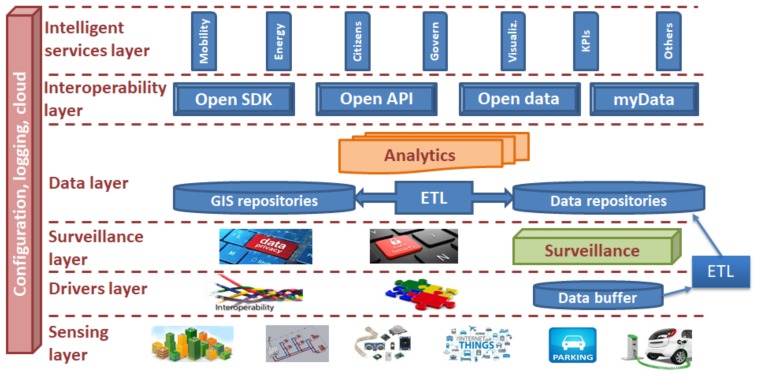
mySMARTLife proposed Open Specifications Framework.

**Figure 4 sensors-20-02402-f004:**
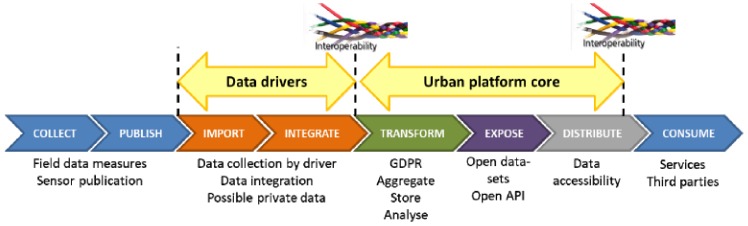
Data management procedure defined within mySMARTLife.

**Figure 5 sensors-20-02402-f005:**
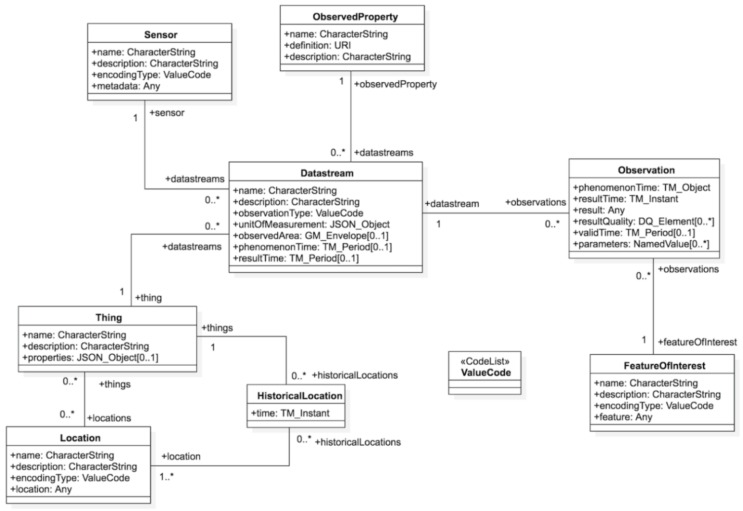
SensorThingsAPI (application programming interface) entity-relationship schema [29].

**Figure 6 sensors-20-02402-f006:**
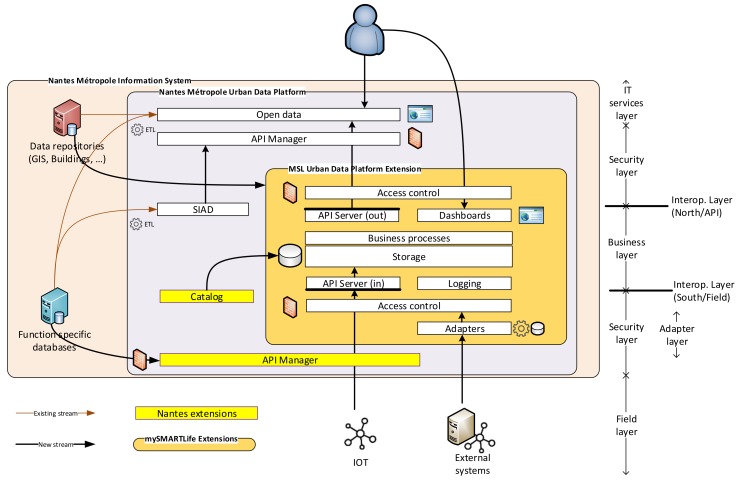
Nantes urban platform reference architecture.

**Figure 7 sensors-20-02402-f007:**
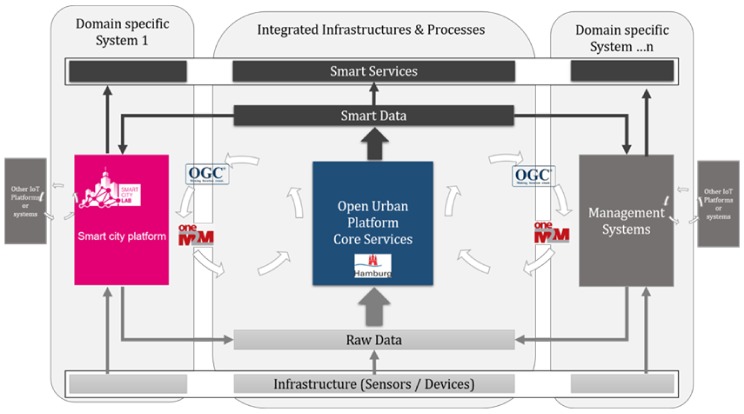
Adapted graphic of DIN Spec 91357 towards an Open Standard-based Urban Platform of Hamburg in mySMARTLife. It follows a “system of systems” according EIP SCC (European Innovation Partnership on Smart Cities and Communities) ensuring full interoperability using standardized APIs and connectors from OGC (Open Geospatial Consortium) and oneM2M.

**Figure 8 sensors-20-02402-f008:**
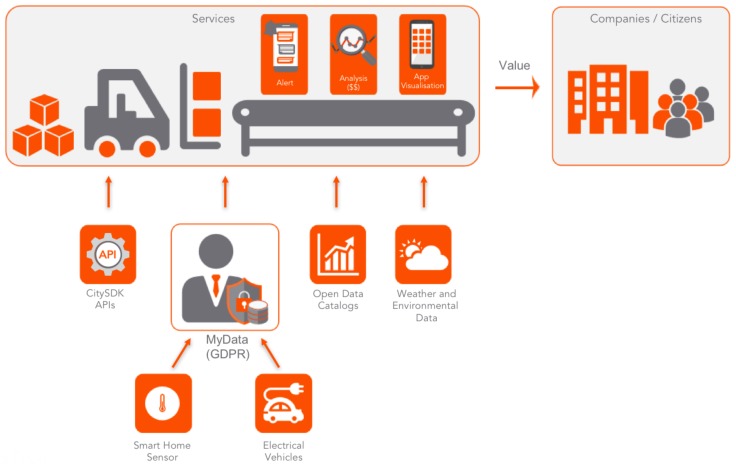
Helsinki urban platform reference architecture under the City-as-a-Platform concept.

**Table 1 sensors-20-02402-t001:** Mapping the Open Specifications Framework and Nantes Urban Platform.

Nantes Layer	Framework Layer	Functionalities
Field layer	Sensing layer	IoT equipment for monitoring, external systems
Adapter layer	Driver layer	Data integration, filtering and transformation.
Field/South and API/North interoperability layers	Interoperability Layer	Data sharing via Open APIs
Business layer	Knowledge layer and interoperability layer	Data aggregation, anonymization, calculation of KPIs, data analysis and data services.
IT services	Intelligent services layer	Visualization and applications to be developed within or outside the project
Security layer	Surveillance layer	Access and GDPR aspects
Common services	Configuration, logging and cloud	Configuration, logging, job control

**Table 2 sensors-20-02402-t002:** Mapping the Open Specifications Framework and Hamburg Urban Platform.

Hamburg Layer	Framework Layer	Functionalities
System and Field component	Sensing and driver layer	IoT devices for data gathering and ingestion into the urban platform.
Integration layer	Knowledge and interoperability layer	Data management and analytics calculation via indicators and big-data. APIs and data sharing.
IT services	Intelligent services layer	Dashboards, added-value services, 3rd party apps…
Security layer	Surveillance layer	Anonymization and GDPR aspects

**Table 3 sensors-20-02402-t003:** Mapping the Open Specifications Framework and Helsinki Urban Platform.

Helsinki Layer	Framework Layer	Functionalities
City backend	Sensing and driver layer	Data ingestion, integration and governance
Events, analysis, ETL	Knowledge layer	Data management and indicators
SmartCity API	Interoperability layer	Open API, Data portals and SDKs
Dashboard, city BI, apps	Intelligent services layer	Dashboards, added-value services, 3rd party apps…
Authentication, MyData console	Configuration, logging and cloud	Access rights and configuration of the urban platform

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
