# Peer review of "Interoperable Open Specifications Framework for the Implementation of Standardized Urban Platformsâ€"

_sensors, 2020, doi:10.3390/s20082402_

Round 1
Reviewer 1 Report
Generally speaking, the paper is suitable for Sensors, I recommend the revision, following are some suggestions. 1.In the literature review section, the smart service and cities are not well introduced, the authors should consider more papers related to smart cities, especially published by Sensors, such as the follows. Chaturvedi, K. and Kolbe, T.H., 2019. Towards establishing cross-platform interoperability for sensors in smart cities. Sensors, 19(3), p.562. Espinoza-Arias, P., Poveda-Villalón, M., GarcÃa-Castro, R. and Corcho, O., 2019. Ontological representation of smart city data: from devices to cities. Applied Sciences, 9(1), p.32. Chen, X., Wang, H.H. and Tian, B., 2019. Visualization model of big data based on self-organizing feature map neural network and graphic theory for smart cities. Cluster Computing, 22(6), pp.13293-13305. Garcia Alvarez, M., Morales, J. and Kraak, M.J., 2019. Integration and exploitation of sensor data in smart cities through event-driven applications. Sensors, 19(6), p.1372. 2.Existing initiatives at European level should be discussed more clearly, for example, the background and the further technique information. 3.The figure 1 is not well discussed, the authors should highlight each component in the figure and make clear discussions. 4.More discussions on the novelty of sensors should be indluded. 5.Discuss the table 1 in detail. 6.The experiment section should be highlighted, and the future work should be added in the conclusion section.Author Response
Dear reviewer,
I would like to explain the changes included according to the review provided by you.
- Literature review section
- This section has been modified in order to include, on one hand, the European initiatives and, on the other hand, research studies have been analysed.
- Existing initiatives at European level should be discussed more clearly
- As described above, the literature section has been modified, where the discusion has been improved as well, not only with respect to the European initiatives, but also compared to the researches.
- The figure 1 is not well discussed
- This has been removed as this is not the important part of the paper, explanations about the initiative has been kept, but without providing the details about all the layers, which is out of the scope of the paper.
- More discussions on the novelty of sensors should be indluded.
- Literature section has been extended and improved with more discusions about the novelty of the Open Specifications Framework presented.
- Discuss the table 1 in detail.
- Table 1, 2 and 3 have been better described and improved with the implementation of the framework on each city.
- The experiment section should be highlighted
- Nantes, Hamburg and Helsinki sections have been extended and further detailed, while the conclusions section already inclueded future work that is being done under the project umbrella.
Thanks
Reviewer 2 Report
This paper proposes a framework presenting four main characteristics, to know, data interoperability, services or verticals interoperability, openness, and replicability. Besides, this study suggests the implementation and integration of multiple city domains for smart management and Big Data analytics.
At first, the manuscript needs a careful writing review. It is possible to find lot of minor errors needing correctness.
For this reviewer, the introduction does not present a deep backgroud on the topic. Important research is neglected, for example:
Smart cities: A survey on data management, security, and enabling technologies. IEEE Communications Surveys & Tutorials, v. 19, n. 4, p. 2456-2501, 2017. DOI: 10.1109 / COMST.2017.2736886
ISO-standardized smart city platform architecture and dashboard. IEEE Pervasive Computing, v. 16, n. 2, p. 35-43, 2017. DOI: 10.1109 / MPRV.2017.31
About intelligent frameworks:
A comprehensive review on smart decision support systems for health care. IEEE Systems Journal, v. 13, n. 3, p. 3536-3545, 2019. DOI: 10.1109 / JSYST.2018.2890121
A comparative table could significantly improve the presentation of initiatives at the European level in Section 2.
The authors need to answer the following question: what is the main contribution of the proposed framework in relation to those presented in Figures 1 and 2?
Improve the quality of Figure 5 for a better understanding.
The authors must include the main limitations of the proposal.
Author Response
Dear reviewer,
I would like to explain the changes made in the paper trying to answer your comments:
- The manuscript needs a careful writing review
- English writing has been reviewed
- The introduction does not present a deep backgroud on the topic
- Literature section has been improved in order to introduce more research articles, which have been also discussed in contrast to the mySMARTLife results.
- A comparative table could significantly improve the presentation of initiatives at the European level in Section 2.
- This table has not been created due to the quantity of initiatives, projects and researches, but, on the contrary, the contributions from mySMARTLife has been cleared describe.
- what is the main contribution of the proposed framework in relation to those presented in Figures 1 and 2?
- Literature section has been significantly improved in order to make clearer the contributions from mySMARTLife not only with respect to the Figure 1 and Figure 2. but also with respect to the new researches that have been included.
- Improve the quality of Figure 5 for a better understanding.
- figures in all the sections related to the cities have been improved.
- The authors must include the main limitations of the proposal.
- The discussion section has been extended with the limitations that the proposal has.
Thanks
Reviewer 3 Report
The study “Interoperable Open Specifications Framework for the implementation of Standardized Urban Platforms” is an interesting issue for data development and sharing aspect. Some of the methodology parts, result and discussion are still not clear. So authors need to elaborate more about the significance of research and how their findings support the policy makers and planners and all smart cities in developed and developing countries. So introduction part, methodology and result section need to improve in the manuscript. The manuscript need to correct missed references, syntax error, and justify the topic sentences.
Major Revision
Need to describe the standard of Smart cities and its paradigm in globally. Now the concept of smart cities are applying in developing countries also. So try to review here more about the standard of smart cities provided by European Commission (ref. 2) and compare it. Need to review more about other data sharing model its capabilities and flexibility and more description about gap analysis.
Need to describe more about mySMARTLife Open Specifications Framework and others with its flow (Figure 2, 3, 4) and verify the figure number based on text (Figure 3,4 and 5….)
Figure1. ITU-T reference architecture for Smart City Urban Platforms (Prepare high resolution figure with large scale and insert in supplementary file)
Figure 5. Sensor ThingsAPI entity-relationship schema Prepare high resolution figure with large scale and insert in supplementary file)
Figure7. Hamburg urban platform reference architecture Prepare high resolution figure with large scale and insert in supplementary file)
Change, figure 8 Helsinki urban platform reference architecture make it high resolution and verify table 3 also.
Authors need to describe more about discussion part with more references.
Line 450- insert reference.
Author Response
Dear reviewer,
I would like to describe the changes and modifications in the paper trying to deal with your comments.
- Some of the methodology parts, result and discussion are still not clear
- Introduction, background, discusion and conclusions sections have been improved in order to include the methodology that has been followed (intro section), discusion includes the benefits and limitations. Finally, English and correctness of the writing has been reviewed.
- Need to describe the standard of Smart cities and its paradigm in globally.
- Literature section has been extended with more analysis and other research results with the aim of providing a better overview about the standard of Smart Cities from multiple perspectives and points of view.
- Need to describe more about mySMARTLife Open Specifications Framework
- Not only the Framework has been better described, but the interoperability issues and its implementation in the cities.
- Figures resolution
- Figures have been replaced to provide other with higher quality.
- Authors need to describe more about discussion part with more references.
- Literature section has been extended with more references and more studies.
Thanks
Round 2
Reviewer 1 Report
For the revised paper, the authors have replied to the reviewers' comment well, the acceptance can be considered.
Author Response
Many thanks for your reply.
Reviewer 3 Report
Dear Authors, I enjoyed reading this revised version of your manuscript. Since, some of my major comments are still not response, mainly in figure. I have some minor comments, and authors need to address this comments before acceptance for the publication.
The details affiliation address is missing of all authors.
Line 127-144 ( Please confirm this order).
Line 203-217, Please review more published scientific work and add more reference here.
Figure number 1 is not visible in my PDF , please make it visible at least A4 size.
Figure 5 also not display (not readable) properly, need to make it again .
Need to make correction in missed references, syntax error,
Author Response
Dear reviewer,
We have corrected the paper according to your comments. Below, our explanation about how we have proceed:
- The details affiliation address is missing of all authors.
- All the affiliation details for the authors have been included with the exception of one author that is freelance, although, during the project, he was part of the consortium.
- Line 127-144 ( Please confirm this order).
- Text has been swapped, but also checked and corrected to be clearer in the message.
- Line 203-217, Please review more published scientific work and add more reference here.
- More scientific articles have been added and the section regarding state of the art has been extended with these new research results
- Figure number 1 is not visible in my PDF , please make it visible at least A4 size.
- Figure 1 size has been increased up to the width of the page.
- Figure 5 also not display (not readable) properly, need to make it again.
- Figure 5 modified and included bigger in the text. In order to help the reading, we have spinned it up, as well as increased its size.
- Need to make correction in missed references, syntax error.
- All the references have been checked, some of them edited to be corrected and the references in the text have been also corrected.
Many thank